# Photosynthesis Responses of Tibetan Freshwater Algae *Chlorella vulgaris* to Herbicide Glyphosate

**DOI:** 10.3390/ijerph20010386

**Published:** 2022-12-26

**Authors:** Yixiao Zhang, Zixu Chen, Xiaoyan Li, Xinguo Wu, Lanzhou Chen, Gaohong Wang

**Affiliations:** 1School of Science, Tibet University, Lhasa 850000, China; 2Institute of Hydrobiology, Chinese Academy of Sciences, Wuhan 430072, China; 3School of Resource & Environmental Science, Wuhan University, Wuhan 430072, China

**Keywords:** Tibetan algae, glyphosate, photosynthetic activity, chlorophyll fluorescence, antioxidant system

## Abstract

With the development of agriculture and the widespread application of agrichemicals in Tibet, herbicide residues have become a threat to the ecological safety of Tibetan water bodies. Algae, as the producers in the food chain in water bodies, play an important role in aquatic ecosystems. Therefore, the impact of herbicides on Tibetan algae is of great significance for evaluating ecological health and the protection of Tibetan water ecosystems. In this study, we investigated the inhibitory effect of glyphosate, a herbicide, on the photosynthetic system of *Chlorella vulgaris*, Tibetan algae, by determining chlorophyll fluorescence and the activity of an antioxidant system. The results revealed that glyphosate at low concentration did not affect the photosynthetic activity of *C. vulgaris*; however, glyphosate at a high concentration significantly inhibited photosynthetic activity and reduced pigment content. Moreover, high levels of glyphosate also decreased photochemical efficiency and electron transport rate and resulted in ROS accumulation, high SOD activity, and lipid peroxidation. These results suggested that glyphosate could decrease the primary production of aquatic ecosystems and influence their performance. Therefore, reducing the herbicide levels could protect the Tibetan aquatic environment and maintain the health of ecosystems.

## 1. Introduction

The Qinghai-Tibetan Plateau is considered “Asia’s Water Tower”. It has the headwaters of several large and important rivers in the world [1,2,3]. There are approximately 1/5th of the world’s human population lives along the watersheds that originate in this region. Moreover, Tibet Autonomous Region is the region with the largest number of lakes and waterbodies in China [2,4]. Because it is located on a high-altitude plateau with an extremely harsh and fragile environment, the ecosystem of waterbodies in the Tibet Autonomous Region is vulnerable to external factors and not easy to be recovered when destroyed. Therefore, the study and protection of the aquatic ecosystem in the Tibet Autonomous Region has extremely great scientific significance and ecological service value [3].In recent years, with the development of agriculture in the Tibet Autonomous Region and the rise in labor costs, an increasing number of farmlands have adopted herbicides for the control of farmland weeds [5]. Because of the characteristics of the Tibet Autonomous Region, compared with other regions in the country, this region has the widest area with weeds in farmland; prevention and control of weeds are very difficult, and the region faces ecological harm due to weeds and herbicides to the greatest degree. The number of agrichemicals used in the whole region reached 0.098 million tons, of which herbicides accounted for 46.48% (8.9% higher than the national average) [6]. Among herbicides, glyphosate is one of the most widely used herbicides [7]. At the same time, compared with Taihu Lake, with a similar high herbicide consumption, the content of glyphosate in Taihu Lake (Jiangsu, China) water can be as high as 52,052 μg/L (0.3 Mm/L), with an average concentration of 6769 μg/L [8,9]. Moreover, Glyphosate can inhibit the production of the enzyme EPSP (5-enolpyruvylshikimate-3-phosphate) synthase and prevents plants from forming the aromatic amino acids that are components of many plant pigments and proteins so that plant growth is disturbed and the plant dies [10,11]. Herbicides such as glyphosate eventually flow into water bodies with rainfall, affecting primary producers of aquatic ecosystems and affecting the health and stability of ecosystems, and glyphosate residues can cause harm to non-target aquatic organisms [12,13]. Studies also have shown that glyphosate exposure can lead to various health hazards, including liver and kidney toxicity, neurotoxicity, and even carcinogenicity, and Camacho found that miscarriages and dermatological and respiratory illnesses are related to glyphosate exposure during aerial glyphosate spraying campaigns to eliminate coca plants in Colombia [14,15]. However, at present, there is a lack of research on glyphosate in water bodies, especially in Tibetan area, and no study has reported the impact of herbicides on the algae species endemic to Tibet’s water body, and this investigation is urgent in the view of the state’s demand that the western region should not engage in large-scale development and carry out a national strategy of large-scale protection.

Algae are the primary producers in water bodies and play essential roles in aquatic ecosystems, and green algae are one of the dominant algae species in the water bodies of many places in Tibet. As an important primary producer in aquatic ecosystems, algae play an important role in both freshwater and marine food chains [16,17,18] and are easily the starting point for pollutant enrichment in the food chain. Therefore, the impact of herbicides on algae in Tibet is of great significance for assessing ecological health and the protection of the water environment. At the same time, it is possible that herbicides absorbed by aquatic algae could be transferred along the food chain to affect human health. Since photosynthesis is the most important biological process in algae, the major investigation on the impact of herbicides on algae mainly focuses on photosynthesis. In recent years, the rapid chlorophyll fluorescence kinetic curve (OJIP test) has been considered an essential technical means to study the photosynthesis process and is widely used in studying environmental stress, ecotoxicology, and ecological adaptation [19,20,21,22]. Rapid chlorophyll fluorescence kinetic curves are becoming increasingly important research tools for biologists because they can record the fluorescence change process from the O point to the P point in detail, fully reflecting the detailed information of the photosynthesis process [23,24,25]. In this study, we investigate the biological mechanism of glyphosate, underlying its impact on the photosynthesis of *Chlorella* in water bodies in Tibet, using rapid chlorophyll fluorescence kinetic curve technology combined with physiological determination. This study provided an evaluation of the health of the ecological environment of water bodies in Tibet and insights into the protection of the water ecological environment.

## 2. Materials and Methods

### 2.1. Algal Culture and Herbicide Treatment

The experimental strain *Chlorella vulgaris* FACHB-2723 was provided by Freshwater Algae collections of Institute of Hydrobiology (FACHB), Chinese Academy of Sciences. The algae strain was originally isolated from Niyang River in Gongbujiangda in Tibet Autonomous Region, China. The alga was cultured in BG11 medium at 25 °C and 35 μmol/m^2^/s light with 12 h light and 12 h dark cycle [26,27,28,29]. Algal in the logarithmic growth phase were collected and exposed to BG11 medium with various volume glyphosate (Aokun Crops Co. Ltd., Shandong, China). The final volume of the culture medium was 20 mL, and the final concentrations were 0, 1, 10, 20, and 40 mmol/L.dium [29,30,31]. Samples were harvested after exposing experiment for further analysis at various time intervals (1, 2, 4, 8, and 12 h) [32].

### 2.2. Determination of Chlorophyll a Fluorescence

The fluorescence of chlorophyll a was determined by a portable plant efficiency analyzer (Handy-PEA, Hanstech, UK). The algal cultures were kept in dark for at least 10 min before measuring the fluorescence parameter Fv/Fm (PSII activity). The excitation light intensity was approximately 3000 µmol m^−2^ × s^−1^, and the recording time was 5 s. The fluorescence intensities at 50 ms, 2 ms (J-step), 30 ms (I-step), and 1 s (P-step) were denoted as FO, FJ, FI, and FP of the OJIP-test, respectively. Analysis of the OJIP test was conducted on the program that comes with the equipment to obtain relevant parameter data [25].

### 2.3. Evaluation of Photosynthetic Pigments

Chlorophyll a, Chlorophyll b, and carotenoids were extracted with 80% (*v*/*v*) acetone, and their levels were determined according to the method described by Wellburn [33]. In brief, 2 mL algal cultures were centrifuged at 2000 rpm speed, then the superior liquid was discarded, and the samples were added 80% (*v*/*v*) acetone and placed at 40 °C for 24 h. Light absorption values of superior liquid were measured with UV-Photometer (7U-1810) at 663, 646, and 470 nm. The contents of chlorophyll a (Ca), chlorophyll b (Cb), and carotenoid (Cc) were calculated using the following Formula (1).
Ca = 12.21A663 − 2.81A646
Cb = 20.13A646 − 5.03A663
Cc = (1000A470 − 3.27Ca − 104Cb)/229(1)

### 2.4. Determination of ROS, SOD Activity and MDA and Protein Contents

ROS was measured using Beyotime kits as per the manufacturer’s instructions using the DCFH-DA method. In brief, 10 mL of algal solution was eluted three times with PBS. The cells were suspended in diluted DCFH-DA, incubated at 37 °C for 20 min, and mixed upside down every 4 min. The fluorescence intensity (488 nm/525 nm) was detected using a Multi-Mode Microplate reader (Filter Max F5; Molecular Devices). The relative value of ROS fluorescence intensity was obtained using the factor fluorescence value/OD730 [34].

SOD activity and MDA contents were determined using Nanjing Jiancheng kits as per the manufacturer’s instructions [35]. In brief, 20 mL of the algal solution was washed three times with PBS. Further, 10% homogenizing medium was prepared by adding PBS at a ratio of 1:9 and grinding (the 20 s, 5 times, interval 20 s). Finally, the suspensions were centrifuged at 5000 rpm for 5 min, and the enzymatic extracts were obtained from the supernatant. The determination of MDA and SOD was carried out according to the instructions.

### 2.5. Statistical Analysis

We used one-way analysis of variance (SPSS-13 for Windows; tests: Duncan’s post hoc test) for data analyses. All experiments involved three technical repeats and three biological repeats.

## 3. Results

### 3.1. The Effects of Glyphosate on Thephotosynthetic Activity

As a photosynthetic organism, the photosynthetic activity of *C. vulgaris* is affected by herbicides. Figure 1 shows that different concentrations of glyphosate had different effects on algae photosynthetic activity. Compared with the control group, the low concentration of glyphosate (1 mM) had little effect on the photosynthetic activity of algae which had no significant effect on the photosynthetic activity. Medium concentrations of glyphosate (10 and 20 mM) exhibited significant inhibitory effects on the photosynthetic activity, and its inhibition effect increased with concentration and treatment time. The high concentration of glyphosate (40 mM) has the greatest inhibition effect on the photosynthetic activity of *C. vulgaris*; even within 1 h of treatment, the photosynthetic activity was reduced by 95%, and the photosynthesis was completely inhibited after treatment for 4 h.

### 3.2. The Effects of Glyphosate on the Contents of Photosynthetic Pigments

Chlorophyll a, as the central pigment of the photosynthetic system, plays an important role in the light reaction in the photosynthetic process, so it is necessary to study the effect of the herbicide glyphosate on the chlorophyll a content. Figure 2 showed that low concentrations of glyphosate (1 mM) had little effect on chlorophyll a content in algae, with no significant differences compared to the control. Medium concentrations of glyphosate (10 mM and 20 mM) decreased the chlorophyll a content of algae, and their inhibitory effect increased with concentration. The high concentration of glyphosate (40 mM) had the greatest effect on the chlorophyll a content, and its chlorophyll a content was only 1/10th of that of the control samples.

Chlorophyll b, as a photo-harvesting pigment of the algal photosynthetic system, plays an important role in photosynthesis. A low concentration of glyphosate (1 mM) exhibited a little and insignificant effect on chlorophyll b content compared with the control (Figure 2B). The treatment with glyphosate at concentrations higher than 10 mM significantly reduced chlorophyll b content, indicating that glyphosate exhibited a certain inhibitory effect on chlorophyll b synthesis.

As an accessory pigment of the photosynthetic system, carotene assists in light absorption and dissipation of excess light energy. A low concentration of glyphosate (1 mM) exhibited little effect on the carotene content, whereas the treatments with medium concentrations (10 and 20 mM) significantly reduced the carotene content in *C. vulgaris* (Figure 2C). A high concentration of glyphosate (40 mM) had the greatest effect on carotene content, and the carotene content was less than 1/6th of that in the control.

### 3.3. The Effects of Glyphosate on Chl a Fluorescence Induction Kinetics

As shown in Figure 3, glyphosate exhibited a significant effect on the photosynthetic Chl a fluorescence induction kinetics. The fluorescence intensity values of the treatment groups with glyphosate concentrations above 5 mM were lower than those in the control group, and the O, J, I, and P phases of the low-concentration treatment groups (1 mM and 10 mM) remained normal, while the inhibition of photosynthesis in the high concentration treatment group (20 mM and 40 mM) was obvious, and the induction curve lacked intermediate J and I phases, and directly entered the P phase, which indicated that the photosynthetic electron transmission was seriously damaged by a high concentration of glyphosate.

To further study the effect of glyphosate on the photosynthetic electron transport pathway of *C. vulgaris*, we conducted an in-depth analysis of the parameters of the rapid chlorophyll fluorescence induction kinetics under various concentrations of glyphosate. As shown in Table 1, the low concentration of glyphosate (1 mM and 10 mM) has little effect on the relevant parameters of the photosynthetic electron transport chain of *C. vulgaris*, and some parameters even have a certain promoting effect, while the medium and high concentrations (20 mM and 40 mM) have obvious inhibition of electron transport. Specifically, VJ reflects the degree of closure of the active reaction center at 2 ms of illumination, whereas MO reflects the maximum rate at which Q_A_ is reduced; both of these were significantly increased by glyphosate. Electron transfer efficiency can be reflected by the specific energy fluxes (per reaction center, RC) and the ratio of energy fluxes, which demonstrated that after glyphosate treatment, the energy transport per reaction center (ETo/RC) and the ratio of energy for electron transport (φE0) had decreased significantly, and the accumulation of a large amount of absorbed energy had caused ABS/RC to increase. Further, heat dissipation of the light reaction center energy (DIo/RC) and distribution ratio of dissipation (φD0) significantly increased, which indicated that the distribution ratio of energy for heat dissipation increased under glyphosate treatment.

### 3.4. The Effects of Glyphosate on Antioxidant Systems

Glyphosate exhibited a significant effect on the antioxidant systems in *C. vulgaris*. Glyphosate treatment significantly increased the production of ROS in algal cells compared with the control group (Figure 4A). ROS levels did not change much after the treatment with glyphosate at 1 and 10 mM, compared with the control group, but compared with control, the production of ROS was approximately three times high after the treatment with glyphosate at 20 and 40 mM. At the same time, glyphosate treatment also caused an increase in the level of lipid peroxidation (MDA) in the membrane of algae cells (Figure 4B) and exhibited a similar trend of ROS production. This demonstrated that the low concentration of glyphosate (1 and 10 mM) did not change much compared with the control group, but the high concentration of glyphosate (20 and 40 mM) increased significantly. In addition, glyphosate treatment led to an increase in the activity of SOD, an antioxidant enzyme, in algal cells. The SOD activity increased in all glyphosate treatment groups compared with the control (Figure 4C). Glyphosate at 1 and 10 mM exhibited higher SOD activity than that at 20 and 40 mM, and as glyphosate concentration increased, SOD activity gradually reduced.

## 4. Discussion

The rapid chlorophyll fluorescence induction curve is a powerful tool for studying and detecting the effects of environmental stress and pollutants on photosynthesis because it can provide extremely abundant information on photosynthetic electron transport [19,25,36]. As single-cell organisms, algae, especially in their photosynthetic system, are very sensitive to stress and pollutants [37,38]. When subjected to stress and pollutants, photosynthetic chlorophyll fluorescence and electron transport parameters change quickly. OJIP test based on chlorophyll fluorescence information can be used to compare the various processes of electron transport in detail to evaluate the inhibitory action of stress or pollutants on photosynthesis and the site of inhibition, which has important scientific value in the field of environmental ecology and stress physiology [23,39,40]. The Chl a fluorescence induction kinetics includes the following phases: after a sufficient dark-adaptation period, the electron receptors of PSII: QA, QB, and PQ completely lose electrons and are oxidized, which has the greatest ability to accept electrons, that is, it is in a “completely open” state, at which time the fluorescence emitted by the sample after being lighted is minimal and in the original phase “O”. When treating the sample with strong light, the electrons produced by the PSII reaction center after being excited are transmitted to QA through Pheo to reduce it, generating QA-, at this time, because QB cannot receive electrons from QA- in time to cause a large accumulation of QA-, the fluorescence rises rapidly, reflecting the J point. QB is able to accept electrons from QA- to form QB2-, causing Pheo, QA, and QB to fully enter the reduced state. At this stage, the PSII reaction center is completely closed and no longer accepts light quanta; the fluorescence yield is the highest, and a P- point appears. The I point that appears in the process of electron transfer from QA- to QB reflects the heterogeneity of the PQ pool; that is, the fast reduce-type PQ pool is first fully reduced (J-I) during the electron transfer process, and further, the slow reduce-type PQ pool is reduced (I-P) [15,19]. The decline stage mainly reflected the changes in photosynthetic carbon metabolism and a gradual decrease in fluorescence [41].

In this study, we observed that the herbicide glyphosate had an effect on the photosynthetic system of Tibetan *Chlorella*, showing no change with low concentration glyphosate, while inhibition with high concentration. This is consistent with the general pattern of inhibition of biological activity by pollutants [19,42,43]. There was a certain dose-effect relationship between the photosynthetic activity (Fv/Fm) and glyphosate concentration. The OJIP test was used to analyze the relevant parameters of the rapid chlorophyll fluorescence induction curve. It was found that glyphosate inhibited the electron transport of the algae photosynthetic system because TRo/RC, ETo/RC, φp0, and φE0 significantly decreased with the increase in glyphosate concentration. TRo/RC reflects the energy captured by the unit reaction center for reducing Q_A_, and it can be found that under glyphosate treatment, its parameters significantly decreased, indicating that glyphosate has an inhibitory effect on the Pheo to Q_A_ process of the electron transport chain. ETo/RC reflects the energy captured by the unit reaction center for electron transport, and the results show that glyphosate also has a significant inhibitory effect on it, indicating a decrease in the energy used for electron transport [25,36,40]. However, it was observed that V_J_ and M_O_ increased under glyphosate treatment compared with the control. V_J_ reflects the degree of closure of the active reaction center at 2 ms of illumination, and M_O_ reflects the maximum rate at which Q_A_ is reduced. V_J_ and M_O_ were significantly increased after glyphosate treatment, suggesting that electron transport from Q_B_ to PQ may be inhibited, and the inhibition site of glyphosate may be located downstream of Q_A_ in the electron transport chain [20,44]. φE0 reflects the quantum yield used for electron transport, which was also reduced under glyphosate treatment, reflecting the obstruction of electron transport. In addition, φp0 reflects the maximum photochemical efficiency, and the experimental results show that glyphosate treatment significantly reduces the φp0 value, indicating that glyphosate treatment seriously affects the photochemical process in *C. vulgaris*, making it inefficient. The influence of glyphosate on electron transport in algal photosynthesis can be reflected in photosynthetic heat dissipation (DIo/RC, φD0, and ABS/RC). DIo/RC reflects the energy dissipated by the unit reaction center, and the results showed that DIo/RC increased significantly with the increase in glyphosate concentration, indicating that a large amount of energy cannot be used by photochemical reactions because of the inability of electron transfer chain. Therefore, the accumulated energy for heat dissipation is greatly increased [32,40]. φD0 reflects the quantum ratio used for heat dissipation, which also showed a significant increase under glyphosate treatment, and also showed a significant increase in the proportion of heat dissipated energy. ABS/RC reflects the light energy absorbed by the unit reaction center, which was also increased under glyphosate treatment. This demonstrated that in the case of electron transfer obstruction, in addition to a part of the heat dissipation, the reaction center had accumulated a large amount of energy at the same time. These large amounts of energy would degrade chlorophyll a and b and carotene. In addition, glyphosate affects the cellular metabolic process and resynthesis of photosynthetic pigments [10,42]. Therefore, intracellular photosynthetic pigments could not be replenished in time. These results indicated that the inhibitory effect of glyphosate on algal photosynthesis is manifested via interfering with photosynthesis electron transport and reducing the photosynthetic pigment system. A similar phenomenon is also found in the algae under herbicides, salinity, heavy metals, mycotoxin, and UV [22,25,36,40,45,46], which exhibited a reduction in photosynthesis electron transmission efficiency and a significant increase in the proportion of heat dissipation energy. In addition, the plasticizer DEHP also increases the proportion of the energy consumed by heat dissipation [32]. Therefore, all types of adversity generally block the photosynthetic electron transport chain in algae, as demonstrated by the low electron transfer efficiency, low photochemical synthesis efficiency, and high heat dissipation energy ratio. The strong decrease in photosynthetic efficiency was confirmed by transcriptomic investigation of algae (*Fucus virsoides*) under glyphosate treatments; the expression of genes involved in photosynthesis and protein synthesis-related processes decreased by 90–95% [11].

Excess energy can be quenched by photosynthetic pigments; however, still, some high-energy electrons leak from the photosynthetic electron transport chain, leading to ROS generation [36,47]. In this study, low concentrations of glyphosate did not affect ROS production and MDA accumulation. However, high concentration increased ROS production and MDA accumulation significantly, which was consistent with other reports [38]. However, SOD activity exhibited varied patterns after glyphosate treatment. The treatment with low concentrations of glyphosate significantly increased SOD activity [42], but as the concentration of glyphosate increased, the SOD activity gradually reduced. This is not reported in other algae species [38,48], where glyphosate decreased SOD activity. The differences may be because *C. vulgaris* uses SOD for ROS removal, whereas other algae depend on other antioxidants (e.g., GSH, APX, Carotenoid) for ROS removal [38,48]. SOD is the main enzyme removing ROS in *C. vulgaris* cells. Therefore, when its activity is induced after treatment with glyphosate at low concentrations, it can quickly quench ROS. Thus, the membrane lipid can be maintained in good state to support highly efficient photosynthesis. Under high concentration of glyphosate, SOD activity was reduced; therefore, ROS could not be removed in a short time. The accumulated ROS can further cause oxidative damage to cells and lipid peroxidation. So SOD plays an essential role in protecting *Chlorella* cells from ROS [42], especially under low concentrations of glyphosate. Because the algal photosynthetic system is located in the thylakoid membrane, and membrane lipid peroxidation can affect the fluidity of the membrane system, glyphosate also reduces the photosynthetic pigment content [19,36,38,42], and especially chlorophyll a is photosynthesis reaction center pigment, its content will also have a certain effect on photosynthesis efficiency. Therefore, glyphosate damaged the photosynthesis system in *C. vulgaris*, eventually reducing the production of the water body and destroying the ecosystem and environment of the Tibetan water body.

In this study, we did not use raw water of a flowing river, but BG11 as the medium of glyphosate exposure, which may limit the assessment of the effect of glyphosate on algae since exposing the algae to glyphosate directly on a medium is very different from the pollution in an aquatic water environment of a flowing river. Moreover, the herbicides in natural conditions are always combined with other pollution and environmental factors, so it is necessary to study the combined effects of herbicide with other pollution or environmental factors in our future investigation, which maybe need the tools of Box Benkhen or Doehlert for experiment designing. Otherwise, it is necessary to strengthen the use standards of herbicide and optimize the use time so as to reduce the loss of glyphosate in the environment. At the same time, we need to develop and utilize different degradation methods, such as biodegradation. Furthermore, in order to explore the response mechanism of algae to glyphosate stress at the molecular level, gene expression analysis will be fulfilled in the future.

## 5. Conclusions

Glyphosate is slightly soluble in water with a long half-life in the water environment. It will rapidly migrate in aquatic plants and continuously accumulate in the cells, continuously harming the aquatic ecosystem, thus affecting the physiological and metabolic activities of aquatic organisms, and even leading to the death of aquatic organisms [49]. In this study, we observed that the herbicide glyphosate has a significant inhibitory effect on the photosynthesis of freshwater algae in Tibet, which mainly interferes with the photosynthetic electron transport chain. It promoted ROS production, lipid peroxidation, and SOD activity, which in turn, reduced the growth of *C. vulgaris* and biomass accumulation in Tibetan water bodies. Since algae are the main primary producers of aquatic ecosystems, these impacts of glyphosate may affect the health of aquatic ecosystem, thus exhibiting a long-term effect on the environmental health and sustainable development of relatively fragile water bodies on the plateau. Therefore, it is extremely important to take measures to further reduce the use of herbicides to protect the ecological health and safety of Tibetan water bodies.

## Figures and Tables

**Figure 1 ijerph-20-00386-f001:**
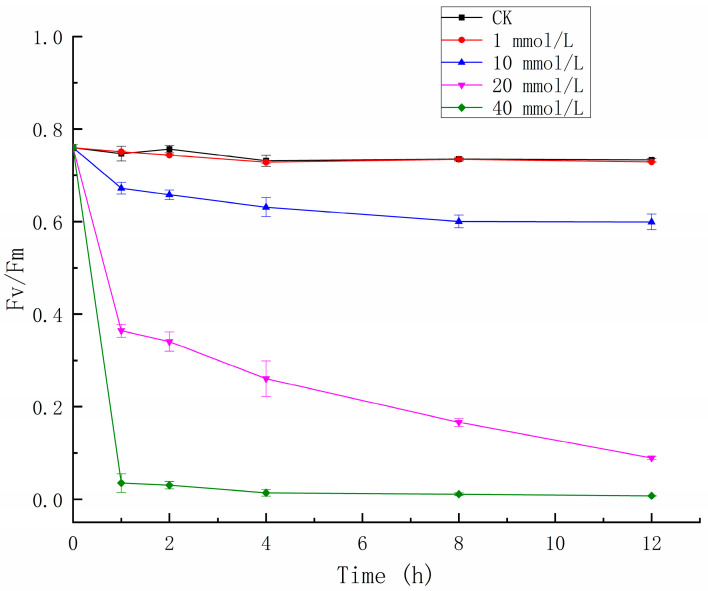
Effects of glyphosate on the photosynthetic activity (Fv/Fm) in *Chlorella vulgaris*.

**Figure 2 ijerph-20-00386-f002:**
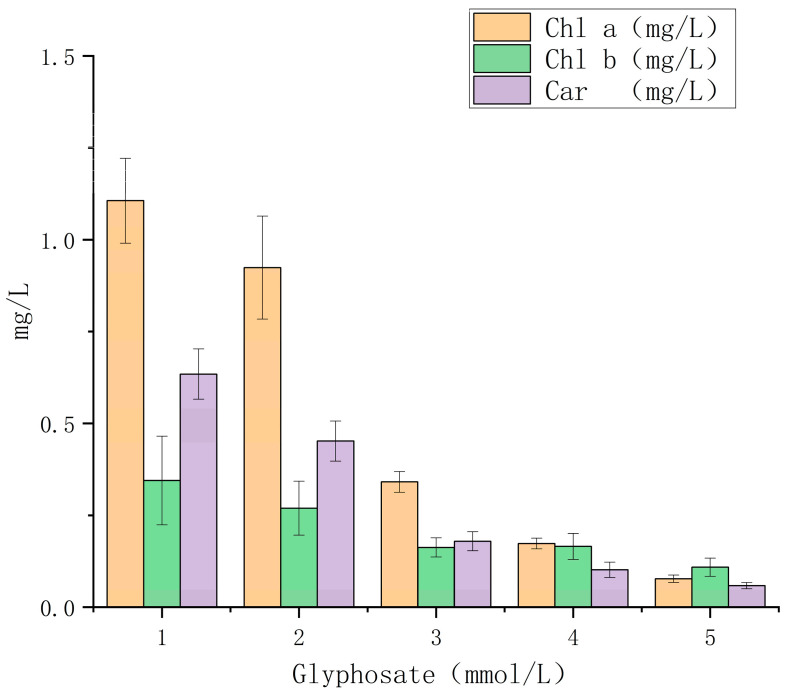
Effects of glyphosate on the photosynthetic pigments of Chlorella vulgaris.

**Figure 3 ijerph-20-00386-f003:**
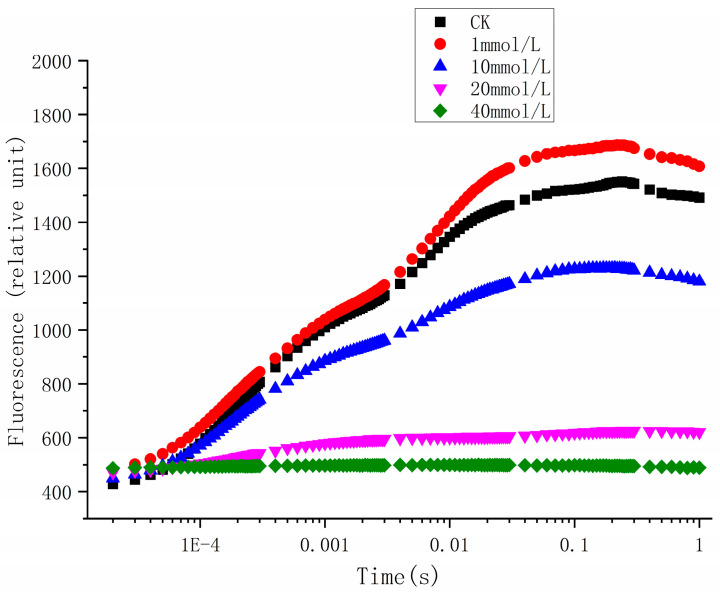
Chl a fluorescence transients in *Chlorella vulgaris* in response to Glyphosate.

**Figure 4 ijerph-20-00386-f004:**
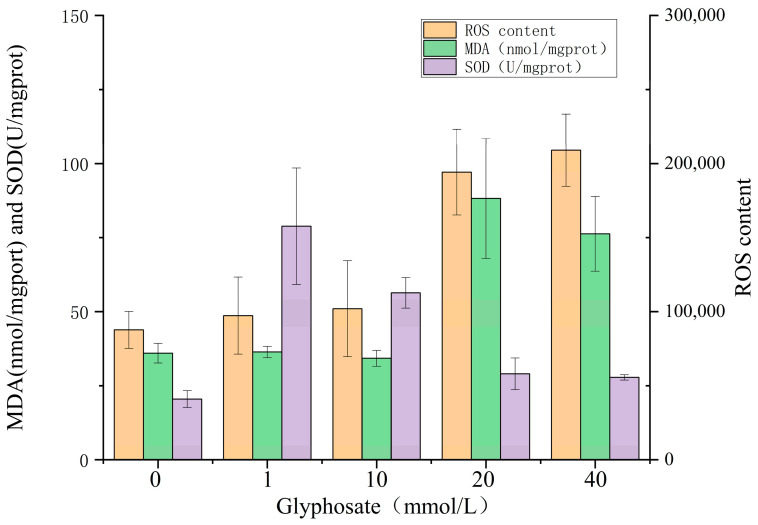
Effects of various concentrations of glyphosate on the antioxidant systems of Chlorella vulgaris.

**Table 1 ijerph-20-00386-t001:** The parameters of Chl a fluorescence induction kinetics under various concentrations of glyphosate in *Chlorella vulgaris*.

Parameters	CK	1 mmol/L	10 mmol/L	20 mmol/L	40 mmol/L
**V_J_**	0.595 ± 0.004	0.528 ± 0.006 *	0.734 ± 0.090 *	0.765 ± 0.012 *	0.956 ± 0.255 *
**Mo**	1.425 ± 0.015	1.228 ± 0.028 *	1.840 ± 0.247 *	1.880 ± 0.021 *	2.915 ± 0.335 *
**ABS/RC**	2.729 ± 0.049	2.663 ± 0.035	3.311 ± 0.175 *	7.954 ± 1.030 *	242.425 ± 128.689 *
**TRo/RC**	1.996 ± 0.008	1.940 ± 0.022	2.088 ± 0.044 *	2.048 ± 0.053 *	2.642 ± 0.668 *
**ETo/RC**	0.808 ± 0.006	0.918 ± 0.003 *	0.611 ± 0.216	0.499 ± 0.025 *	0.569 ± 0.138 *
**DIo/RC**	0.732 ± 0.045	0.723 ± 0.016	1.223 ± 0.134 *	5.906 ± 1.058 *	239.783 ± 128.293 *
**φp0**	0.732 ± 0.012	0.728 ± 0.003	0.631 ± 0.021 *	0.261 ± 0.039 *	0.013 ± 0.008 *
**φE0**	0.296 ± 0.006	0.344 ± 0.005 *	0.169 ± 0.062 *	0.061 ± 0.011 *	0.001 ± 0.003 *
**φD0**	0.268 ± 0.012	0.272 ± 0.003	0.369 ± 0.021 *	0.739 ± 0.039 *	0.987 ± 0.008 *

* *p* < 0.05 compared with the control group (CK).

## Data Availability

The data presented in this study are available within the article.

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
