# Peer review of "Photosynthesis Responses of Tibetan Freshwater Algae *Chlorella vulgaris* to Herbicide Glyphosate"

_ijerph, 2022, doi:10.3390/ijerph20010386_

Round 1
Reviewer 1 Report
General Comments:
Interesting manuscript regarding the role of glyphosate on the photosynthetic capability, pigment abundance and electron activity of a fresh water algae in the Tibetan region. There are some general concerns of the representability of the study and a lack of environmental connection with the findings.
How representative is Chlorella vulgaris in general for the Tibetan water ways, how much is the relative abundance and does the response of this algae correspond with other algae in the same system? These aspects need to be incorporated both in the introduction and the discussion.
The methods (primarily lines 82-84) are unclear, how were the algae exposed to glyphosate, there needs to be specific protocols and procedures to allow for other to replicate the work. Furthermore, and perhaps of a much larger importance, is this exposure representative of their natural environment? Exposing the algae to glyphosate directly on a medium is very different from the pollution in an aquatic water environment of a flowing river. A few sentences must be directed towards this potential study restriction.
Many environmental aspects were overlooked, primarily in relation to the study findings and in three main ways. These needs to be incorporated both in the introduction and discussion.
Firstly: How does glyphosate vary in the representative study area, what types of concentrations are considered ‘natural’ for the area, how is it distributed and are there point areas of pollution or a general leakage into the water ways? What are likely concentrations in the leakage area and what are the concentrations further away from these areas? These aspects are essential in determining how the concentrations in the study are representative for the natural conditions in these systems.
Secondly: How does this affect the food webs? What are the aquatic food webs in these areas and how does glyphosate accumulate? Are some aquatic species more likely to be affected by this?
Lastly: How does the study findings contribute to understanding these natural systems, and are there alternative agrochemicals that can be recommended instead of glyphosate to minimize the damage to these algae, and if so, what other potential issues could this bring?
Specific Comments:
1. Lines 54-55. It is specified here that the species is endemic to the Tibetan region, is this correct?
Figures and Tables:
1. Figure 1. All the figures, but especially figure 1 is in a very poor resolution, it is hardly possible to make out the axes. This needs to be improved.
Author Response
Thank you for your comments, and please see the attachment.

Reviewer 2 Report
The study is important in the context of the need for continued scrutiny of how human activities affect the environment. However, the fatal flaw which I want to draw your attention to is that the actual concentrations in Tibetan surface waters have apparently not been analysed because no such data are presented as the basis for the selection of glyphosate concentrations used in this study. To what do the glyphosate concentrations of 1, 10, 20, and 40 mM relate to in practice? The authors state 1mM is low and 40 mM is high concentration, on what data do they base these assumptions?
Reviewer 3 Report
This paper, entitled Responds of Tibetan freshwater algae Chlorella vulgaris to herbicide Glyphosat, is a scholarly work and can increase knowledge on this domain. The authors provide an interesting and original study, the content is relevant to IJERPH.
I have some general and specific comments:
- The abstract and keywords are meaningful.
- Why focusing on Chlorella vulgaris and not on other strains? Chlorella vulgaris is commonly considered as a model or microalgae or reference, but why especially this one and not another one?
- Why the choosing the medium BG11 for the cultivation? is there any other medium most adapted and easier to prepare? Why not considering raw effluent or raw water for such experiment? I understand that the method requires development, calibration and validation but why not using real medium?
- how were determined the various concentrations of glyphosate for this experimentation? same question for various time intervals?
- about glyphosate, is it commercial product containing glyphosate or pure glyphosate?
- considering the experimental protocol describedin subsection 2.1, why not considering experimental design tool such Box Benkhen or Doehlert for this experiment? It could provide surface response and an optimumn reducing the numbers of experiments.
- why the authors carry out extraction on chlorophyll and carotenoids for meaurement? It could be possible considering spectrophotometric methods to measure directly with spectra and wavelengths, is this method most adapted and more accurate?
- Please improve quality of figures and enlarge the size of these figures.
- The results section is quite descriptive and the authors should discuss more in depth their results, considering also existing literature. The manuscript is quite well writtent but unsufficiently related to existing studies and literature. Please improve this point.
- maybe the histograms for each pigment in Figure 2 could be merged in only one graph, for a better understanding of the results and in order to improve the comparison between all the results.
- Please explain in Figure 3 and in the text related to this figure why the upper curves decrease at the end of the plotted period? What is the phenomenon that could explain this fact? Why some points are lacking? What is the frequency of data acquisition? Please mention that the X axix is logarithmic in the legend.
- considering standard deviation or accuracy of data in Table 1, please check the number of significant digits for all the data.
- About Figure4, samle comment as mentioned for Figure 2 with possibilityto merge all the histograms.
- About discussion and conclusions, what are the recommendations of the authors about the results obtained in this study? What are the next steps and future works?
- Please discuss and provide environmental and health considerations in the discussion and conclusion section. What are the conclusions in terms of environment, public health, agronomy, water quality, ...
As it, this manuscript is not fully acceptable for publication and requires at least some amendments and additional informations. I encourage the authors to consider all the comments and requests of amendments when preparing the revised version of their manuscript if the editors give the chance to revise this work. I recommend the following decision: RECONSIDER AFTER MAJOR REVISION.
Round 2
Reviewer 1 Report
General Comments:
The edits by the authors have improved the manuscript, however one important detail remains uncorrected from the previous feedback. It is imperative that the exposure experiments (lines 90-93) are explained in much more detail. How were the glyphosate added on or in the medium to simulate the natural exposure? Was the glyphosate dissolved in water and then dripped onto the algae? Was it injected directly into the medium using syringes? What was the method and how was it applied?
Reviewer 2 Report
The first word in the title, "Reponds...." strikes me as awkward use of the English language. I propose a minor change in the title: "Photosynthesis responses of ...." or "Photosynthetic responses of ......".
Reference to Taihu Lake (lines 44-46) -- where is this lake located? What glyphosate data are available for glyphosate in the water of this lake? The references provided here, i.e. references no. 8 and 9, do not appear to be relevant for this lake, and neither for Tibetan fresh water lakes.
Based on the presentation of facts, I remain unconvinced that the tested glyphosate dosages have practical relevance.
In my opinion, the very last sentence, under 'Conclusion', "Therefore, it is extremely important to take measures to further reduce the use of herbicides ....." is not supported by the findings of this study.
Author Response
Thank you again for your comments. Please see the attachment.

Reviewer 3 Report
The authors provide a revised version of their manuscript taking into account all the comments and requests of amendments. The authors provide also very detailed and clearly justified answers to all the comments and requests of amendments. I agree with all the answers and I consider thatthe manuscript is now fully acceptable for publication, I recommend the following decision: ACCEPT IN PRESENT FORM.
Author Response
Thank you veyr much for your kind work and consideration, we greatly appreciate.